# Stepwise control of host–guest interaction using a coordination polymer gel

Rahul Dev Mukhopadhyay[1], Gourab Das [1] & Ayyappanpillai Ajayaghosh [1]

Precise control of host–guest interaction as seen in biological processes is difficult to achieve with artificial systems. Herein we have exploited the thermodynamic benefits of a system in equilibrium to achieve controlled stepwise release and capture of cyclodextrin (guest) using a coordination polymer (Mg-CP) as the host and temperature as the stimulus. Since temperature is not a precision stimulus for artificial host–guest interaction, the present system is a distinct prototype that manifests temperature-controlled natural host–guest interaction. The described coordination polymeric host system, when incorporated into a hydrogel matrix, provides a microenvironment that facilitates the stepwise release of $\alpha$-CD in response to temperature variation within a quasi-solid state. The work demonstrated here may pave the way towards thermally controlled delivery and monitoring of otherwise spectroscopically silent molecules such as cyclodextrins.

[1] Photosciences and Photonics Section, Chemical Sciences and Technology Division and Academy of Scientific and Innovative Research (AcSIR), CSIR-National Institute for Interdisciplinary Science and Technology (CSIR-NIIST), Thiruvananthapuram 695019, India. Correspondence and requests for materials should be addressed to A.A. (email: ajayaghosh@niist.res.in)

Ever since the postulation of the lock and key concept by Emil Fischer in 1894[1], a myriad of studies have been reported in the area of host–guest interactions, aiming at understanding the mechanism behind enzymatic action in biological systems. Most of the enzymatic processes such as synthesis of peptides, degradation of various metabolites, synthesis and hydrolysis of adenosine triphosphate, etc. proceed in a stepwise fashion and with absolute control[2]. Several of these natural systems are precisely controlled by external or internal stimuli; however, designing such artificial stimuli-responsive systems remains a challenging task[3]. Owing to its instant delivery at a specific location and control via different wavelengths and intensities, light is arguably the most attractive physical stimulus for artificial systems[4–7]. However, although heat is not considered as a stimulus with precisional control[3,8], natural host–guest processes are mainly controlled by heat. Since equilibrium constants of host–guest interactions are constant at a particular temperature, change of temperature of a system at chemical equilibrium may change the equilibrium constant, followed by the shifting of the equilibrium towards a direction to undo the change according to Le Châtelier's principle[9]. Temperature can therefore take advantage of the thermodynamic benefits of host–guest equilibrium[10,11] and be exploited to demonstrate the formation (capture) or dissociation (release) of a host–guest complex in a stepwise manner. An ideal host system with multiple guests that facilitates temperature-controlled stepwise release–capture process is shown in Fig. 1[12–15]. This strategy may further help us in developing thermoresponsive smart materials[16,17], for example, a drug delivery system similar to the concept of magic bullet propounded by Ehrlich[18], which can be programmed for drug delivery as well as drug re-uptake or clearance from the blood stream essential for tackling issues like drug resistance and side-effects.

Cyclodextrins (CDs) are known to be an excellent host for the host–guest interaction with hydrophobic guest molecules[19]. CDs have therefore been widely exploited as hydrophobic drug-delivery vehicles as well as in the design of functional supramolecular polymers[20,21]. Recently, CDs have also been recognized as potent drug molecules for treating fatal neurodegenerative disorders like Niemann–Pick type C and as anti-obesity dietary fibres for flushing out fat from high cholesterol diets[22,23]. However, a suitable delivery system for easy, detectable and controlled stepwise release of CDs in solution as well as in soft solid-state devices is yet to be realized. This is because, unlike other organic molecules, CDs are optically transparent and their release and binding profiles cannot be identified using common spectroscopic techniques such as absorption or fluorescence. Nevertheless, by

virtue of their inherent chirality, CDs offer the possibility to follow their binding-release isotherms via induced circular dichroism (ICD) spectroscopy[24,25]. For this purpose, a light absorbing moiety possessing accessible binding motifs is required. In this regard, several hydrophobic organic molecules such as azobenzene, adamantane, bipyridine, etc. have been used that bind with CDs[21]. If such moieties can be organized over a one-dimensional (1-D) coordination polymer (CP) (Fig. 1), the system may reversely accept and release CDs in solution upon application of an appropriate stimulus. While CDs are known to be the host in almost all cases, reports on their use as guest are rare[26]. Therefore, the present system is a rare example where CDs behave as the guest in a host–guest molecular interaction.

According to a report by Yaghi and co-workers, **MOF-74**-based structure, composed of a chiral infinite rod-shaped magnesium or zinc oxide cluster and dioxidoterephthalate-based organic linkers, forms a non-interpenetrated framework[27]. The ligand has a carboxylic acid motif with an *ortho*-positioned hydroxy group, as in salicylic acid. The observed structure can probably be restricted along the crystallographic *b* axis, if the organic strut contains the metal-binding functionalities on one end. Such a design leads to a polymer brush-like structure, with the organic linkers organized along the infinite metal oxide chain as reported in the case of infinite CPs of calcium, manganese, cadmium, zirconium, lanthanides, etc.[28]. With this idea in mind, we prepared an organic ligand (**AzSA**) with a hydrophobic azobenzene core and a metal-binding salicylic acid moiety.

Herein we have designed a CP-based host, with azobenzene side appendages, which exhibits temperature responsive host–guest interaction with CDs not only in the solution state but also in the quasi-solid (hydrogel) state. Using the aforesaid system, we have been able to demonstrate a stepwise release and capture profile of CD guests from the CP backbone.

## Results

**Synthesis and characterization of host polymer**. The magnesium-based CP, **Mg-CP** was synthesized via solvothermal reaction of azobenzene salicylic acid, **AzSA**, with Mg(NO$_3$)$_2$·6H$_2$O in high boiling solvent such as *N,N*-diethylformamide (DEF) (Fig. 2a) and characterized using various spectroscopic and analytical techniques (Fig. 2 and Supplementary Fig. 1). Coordination of the carboxylic acid and hydroxyl groups in **AzSA** to the MgO-based inorganic cluster was confirmed by the absence of the free O-H stretching band (2500–3300 cm$^{-1}$) in its Fourier transform infrared (FT-IR) spectrum (Fig. 2b). The unsymmetric nature of the bidentate

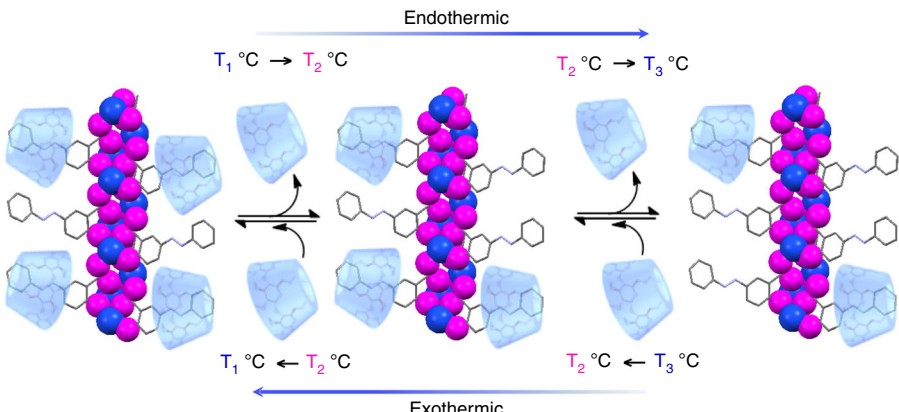

**Fig. 1** Temperature-controlled host–guest systems. Schematic representation of an ideal temperature controlled formation and dissociation of multiple guests (cyclodextrins) from a host (coordination polymer) in a stepwise manner

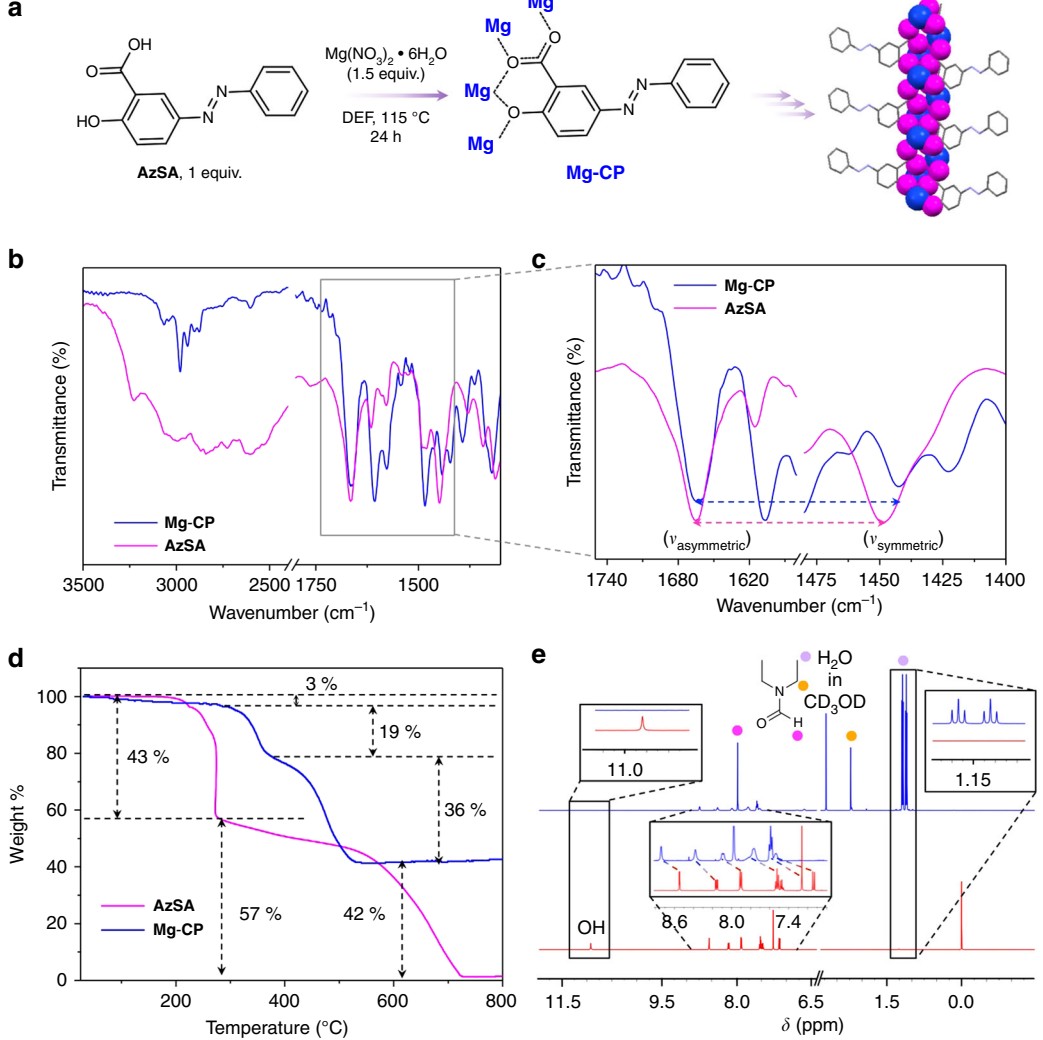

**Fig. 2** Synthesis and characterization of **Mg-CP**. **a** Reagents and conditions for the synthesis of the coordination polymer **Mg-CP** from organic ligand **AzSA**. Schematic representation of coordination modes in **Mg-CP**. Blue and magenta represent magnesium and oxygen atoms, respectively. **b** FT-IR spectra of **AzSA** and **Mg-CP** showing changes in the region corresponding to O-H stretching and the asymmetric and symmetric stretching frequencies of the carbonyl bond. **c** Expanded portion of the region (1750–1400 cm$^{-1}$), showing the differences in the asymmetric and symmetric stretching bands of the carbonyl bond. **d** TGA of **Mg-CP** and **AzSA**. **e** $^1$H NMR spectra of **Mg-CP** (blue) and **AzSA** (red) (500 MHz, CD$_3$OD)

bridging mode of coordination with additional bridging was corroborated by the difference between asymmetric and symmetric stretching frequencies of the carbonyl group that increased from 217 cm$^{-1}$ in **AzSA** to 223 cm$^{-1}$ in **Mg-CP** (Fig. 2c)[29]. Thermogravimetric analysis confirmed the superior thermal stability of **Mg-CP** over the ligand **AzSA**. The thermal decomposition of the 1-D CP was found to start around 300 °C, whereas the ligand (**AzSA**) starts to decompose around 200 °C. Moreover, a residual weight (42%) equivalent to the amount of Mg-O present in **Mg-CP** remained even >700 °C (Fig. 2d). The phenolic –OH proton, which is otherwise exchangeable in CD$_3$OD can be visualized in the $^1$H NMR spectrum of **AzSA** due to strong intramolecular hydrogen bonding. The absence of the phenolic–OH peak in the $^1$H nuclear magnetic resonance (NMR) spectrum of **Mg-CP** confirmed the coordination of the ligand with Mg$^{2+}$. Broadening of the peaks in the aromatic region and the presence of coordinated DEF solvent molecules were also evident from the $^1$H NMR spectrum of **Mg-CP** (Fig. 2e). No changes were observed in the ultraviolet (UV)–visible absorption spectrum of **Mg-CP** even on prolonged 365 nm UV irradiation (Supplementary Fig. 2). There can be two potential reasons

behind this observation; first, the coordination of an azobenzene ligand in a CP or a MOF backbone may lead to the loss of its photoisomerization properties[30–32]. Second, owing to the presence of an electron-donating hydroxy group at the *para* position to the azobenzene functionality, the system may undergo very fast *cis–trans* reverse isomerization[33].

The **Mg-CP** when prepared from DEF resulted in the formation of a CP gel (Fig. 3). The critical gelation concentration was found to be 4 mg mL$^{-1}$ (Fig. 3a; for details, see Supplementary Methods). The gel was found to be thermally irreversible and its non-linear rheological behaviour was confirmed by the amplitude sweep studies (Fig. 3b). The xerogel obtained from **Mg-CP** was further characterized by FT-IR and $^1$H NMR spectroscopy. Wide-angle X-ray scattering (WAXS) analysis (Fig. 3c) of the **Mg-CP** xerogel powder revealed intense diffraction peaks indicating a highly crystalline columnar hexagonal (Col$_h$) structure, with $d$-spacing ratios of $d$, $d/\sqrt{3}$, $d/\sqrt{4}$, $d/\sqrt{7}$ and $d/\sqrt{9}$. From the first diffraction peak of **Mg-CP** ($d_{100} = 19.45$ Å), the hexagonal lattice parameter $a$, was estimated as 22.45 Å, which in turn provides the information about the diameter of the columnar stacks in **Mg-CP** xerogel powder

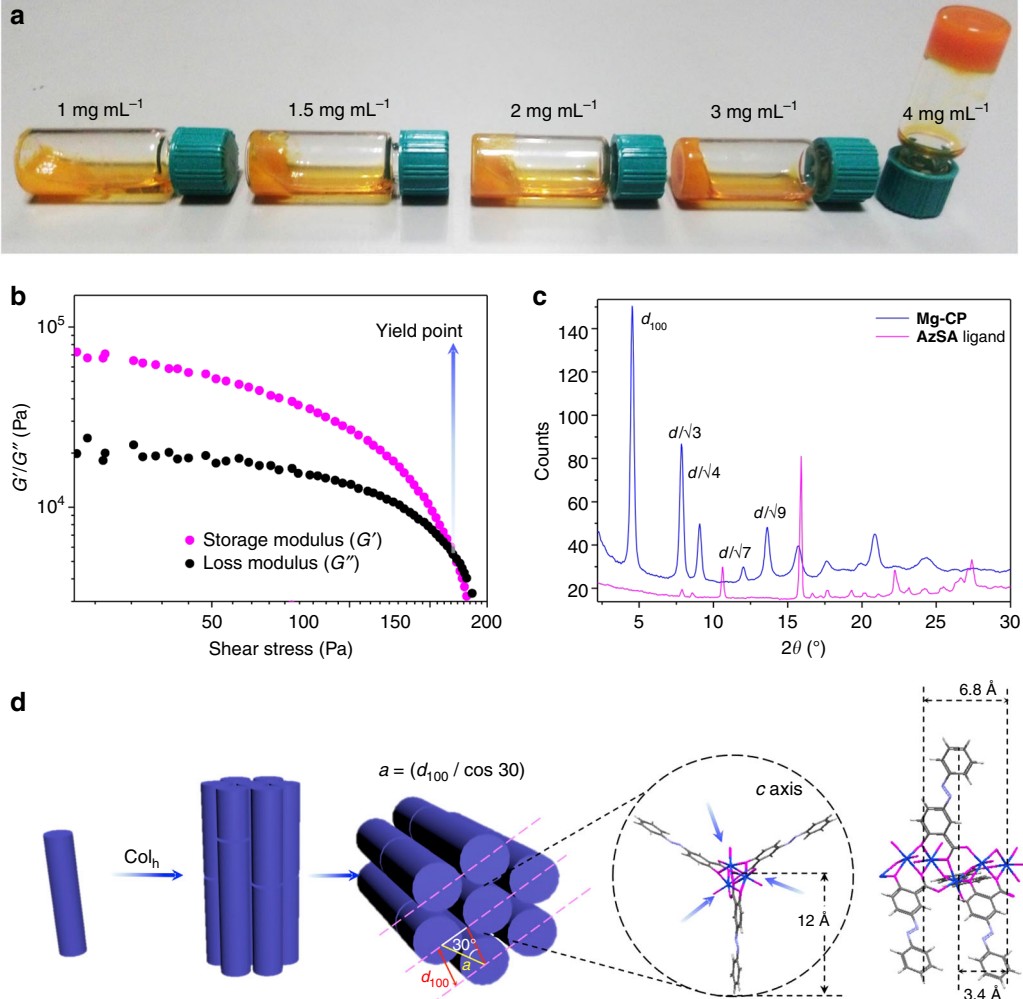

**Fig. 3** Gelation behaviour of **Mg-CP** and structural analysis of the xerogel. **a** Photographs showing the gelation of **Mg-CP** at the critical gelation concentration (4 mg mL$^{-1}$). **b** Non-linear rheological response (amplitude sweep) of **Mg-CP** gel. **c** WAXS analysis of **Mg-CP** and **AzSA**. **d** Schematic representation of the molecular arrangement of columnar hexagonal (Col$_h$) packing of **Mg-CP** polymeric strands and the calculated lattice parameter. Blue arrows represent coordinated solvent molecules

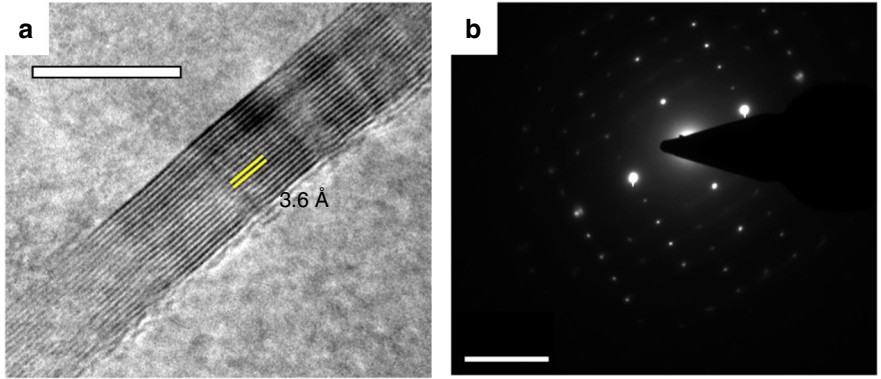

**Fig. 4** Transmission electron microscopic analysis of **Mg-CP** fibres. **a** High-resolution TEM image of **Mg-CP** fibres. Scale bar, 10 nm. **b** Single-crystalline SAED pattern obtained from the agglomerates of **Mg-CP** fibres. Scale bar, 5 nm$^{-1}$

(Fig. 3d)[34]. The diameter determined from WAXS is in good agreement with the distance of ~24 Å, reported for Zn$^{2+}$-osala-zine-based MOF [**Zn$_2$(Olz)**] (Fig. 3d), which is isostructural to its magnesium anlogue[35]. The distances between the parallel and antiparallel organic struts of **AzSA** were estimated to be 6.8 and 3.4 Å, respectively, by comparing with the crystal structure of

**Zn$_2$(Olz)** (Fig. 3d). Whereas the WAXS pattern of **AzSA** matches well with a layered structure similar to its ammonium salt (Supplementary Fig. 3). Transmission electron microscopic images of **Mg-CP** fibres (Fig. 4a and Supplementary Fig. 4) revealed highly crystalline nature of a closely packed and ordered ($d = 3.6$ Å) structure corresponding to a columnar hexagonal

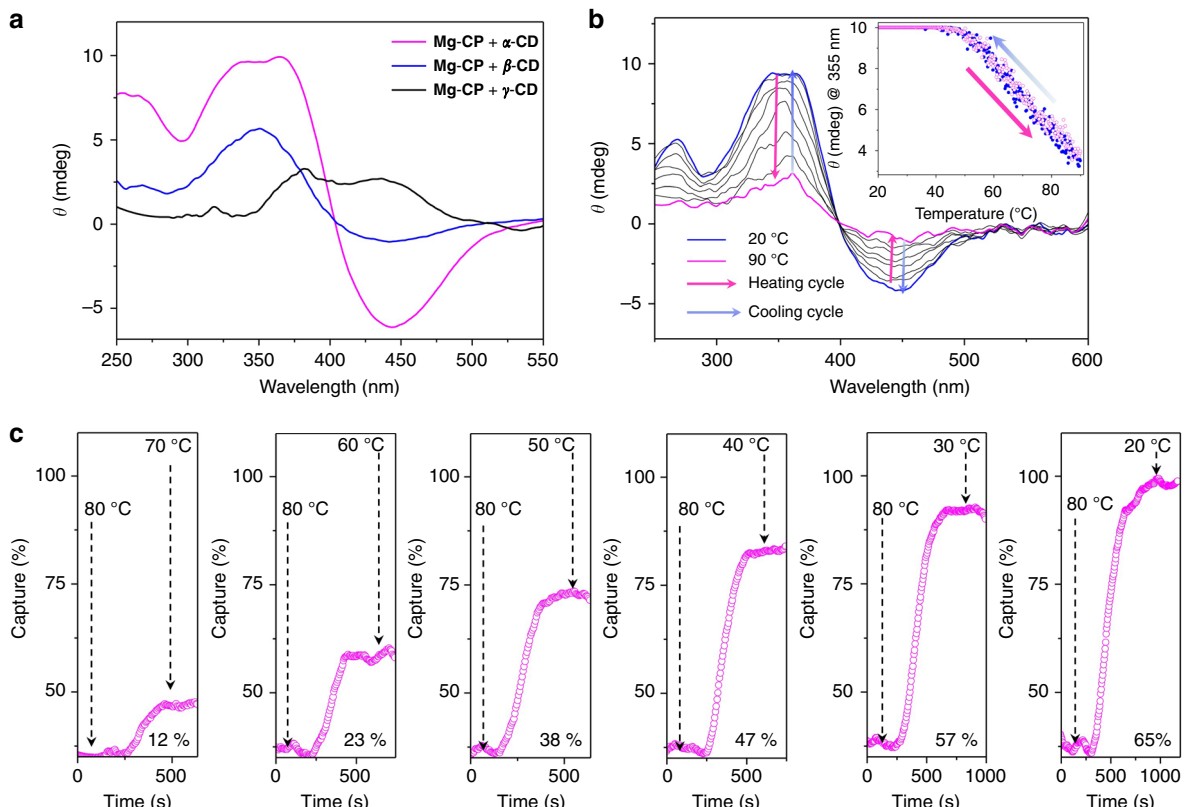

**Fig. 5** Induced circular dichroism (ICD) capture–release studies of CD complexes of **Mg-CP**. **a** ICD spectra of **Mg-CP** in presence of *α*-, *β*- and *γ*-CDs. **b** Temperature-dependent ICD spectra of **Mg-CP⊃α-CD**. Inset shows the changes in the intensity of the ICD band at 355 nm on heating (magenta) and cooling (blue) a solution of **Mg-CP⊃α-CD** in water between 20 and 90 °C. **c** Stepwise temperature-controlled quantitative capture of *α*-CD by an aqueous solution of **Mg-CP**. Concentration of **Mg-CP** is 0.06 mg mL$^{-1}$ and that of *α*-, *β*- and *γ*-CD is 1 × 10$^{-3}$ M

lattice as looked along the *b* axis (Fig. 3d). A selected area electron diffraction pattern corroborating the single crystalline nature was obtained from the larger agglomerates of **Mg-CP** fibres (Fig. 4b).

**Host–guest interaction with cyclodextrins in solution**. The host–guest interaction of the CDs with **Mg-CP** as well as its organic precursor **AzSA** (Supplementary Figs 5-7) was studied in solution state. The intrinsic chirality of CDs allowed us to monitor the binding interactions using a circular dichroism spectrophotometer. A positive ICD band at 355 nm corresponding to the π–π* transition and a small negative band at 440 nm corresponding to n–π* transition (Fig. 5a) indicate the binding of *α*-cyclodextrin (*α*-CD) with **Mg-CP** and **AzSA**. The hydrophobic azobenzene core resides inside the CD cavity with its electronic transition moment parallel to the CD axis[36]. The intensity of the ICD signals at 355 nm was found to be two-fold higher for **Mg-CP** as compared to **AzSA** when titrated with 3.5 and 4.5 equivalents of *α*-CD, respectively (Supplementary Fig. 6a,b). A detailed titration plot of the two systems shows that the ICD response was more consistent in the case of **Mg-CP** as compared to **AzSA** (Supplementary Figs 6c and 7). The ICD spectrum of **Mg-CP⊃β-CD** was similar to that of **Mg-CP⊃α-CD**, suggesting similar orientation of **AzSA** stalks inside the *β*-CD cavity. The interaction of *γ*-CD with **Mg-CP** and **AzSA** was found to be rather weak. The ICD spectrum of **Mg-CP⊃γ-CD** was recorded at a temperature below the room temperature (15 °C) and consisted of a broad positive signal in the range of 350–500 nm (Fig. 5a).

On increasing the temperature of an aqueous solution of *α*-CD encapsulated **Mg-CP** (**Mg-CP⊃α-CD**) in the range 20–90 °C, a steady decrease in the intensity of the ICD signal at 355 and 440 nm was observed. This decrease in intensity correlates to the release of the *α*-CD from **Mg-CP** via temperature-induced decomplexation (Fig. 5b). The release and capture profiles of *α*-CD from **Mg-CP** followed identical pathways and no hysteresis was observed during heating or cooling cycles (inset, Fig. 5b). Similar thermally controlled host–guest complexation was obtained for the *α*-CD complex of **AzSA** (**AzSA⊃α-CD**) (Supplementary Fig. 8). The thermal stability of both the inclusion complexes was determined by monitoring the decrease in intensity of the absorption band at 350/355 nm with respect to the increase in temperature (Supplementary Fig. 6d). The **Mg-CP⊃α-CD** was found to be thermally more stable than **AzSA⊃α-CD**, which is in agreement with our previous observations (Supplementary Fig. 6a, b). **Mg-CP⊃α-CD** was therefore chosen as the model system for demonstrating temperature-controlled capture and release of CDs from the host network. By changing the temperature of **Mg-CP⊃α-CD** solution, the amount of *α*-CD captured or released could be quantitatively controlled. A temperature dependent capture profile is shown in Fig. 5c.

**Thermally controlled host–guest interaction**. One of the most relevant properties of any ideal remotely controlled systems is to dodge the process of point capture or point release. For instance, a high dosage of a drug is often administered at an initial point of treatment and may be repeated at a later stage or the dosage may be lowered after several hours or days. This drug administration protocol cannot be precisely controlled and is not free from side effects. In order to address this problem, scientists have tried to develop efficient methods for continuous and controlled drug

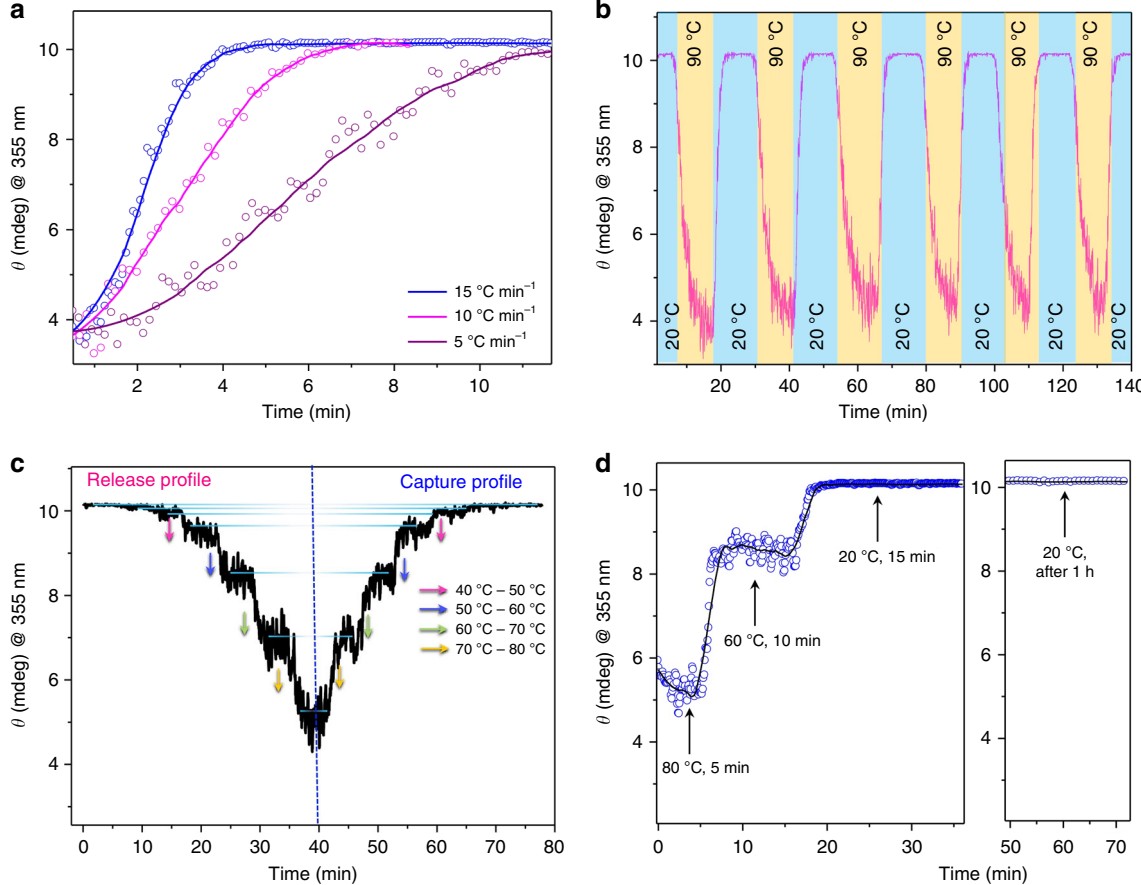

**Fig. 6** Stepwise and reversibly controlled release and capture of **α-CD** by **Mg-CP** in solution state. **a** Temperature-dependent binding of **α-CD** by **Mg-CP** with cooling rate of the solution from 15 to 5 °C min$^{-1}$. Non-linear fit to the observed data points are also represented. **b** Thermally reversible binding (blue) and release (yellow) of **α-CD** by **Mg-CP**. **c** Stepwise release and capture of **α-CD** by **Mg-CP**. The arrows represent the specific changes in the temperature. **d** A two-step capture profile of **α-CD** from **Mg-CP**. The ICD signal remains stable at a particular temperature for the desired amount of time. The plot on the right shows no release of **α-CD** from **Mg-CP** at 20°C even after 1 h. Concentration of **Mg-CP** is 0.06 mg mL$^{-1}$ and that of **α-CD** is 1×10$^{-3}$ M for all experiments

delivery for prolonged time periods or throughout the whole course of the treatment[37]. Most of the stimuli-responsive drug delivery systems reported in literature does not show any capture or release of drug molecules in absence of a stimulus. A gradual and continuous release or capture process starting at a particular point is generally observed upon the application of a particular stimulus, which in most cases cannot be further delayed, controlled or reversed[5].

It has not yet been demonstrated that temperature-dependent capture–release processes can be achieved in a stepwise and reversible manner, with specific control over the amount of guest molecules captured or released. By controlling the cooling rate from 15 to 5 °C min$^{-1}$, we demonstrate that the binding rates of **α-CD** to **Mg-CP** could be controlled from rapid to moderate to slow rates (Fig. 6a). The system also showed reversible binding and release of **α-CD** from the CP over several heating–cooling cycles (Fig. 6b). In line with the conventions of a system in thermodynamic equilibrium, the equilibrium constant or the ratio of the **α-CD** encapsulated sites to that of the free sites present in **Mg-CP** and the amount of free **α-CD** molecules present in solution remains constant at a particular temperature. Further, the ratio of bound to free **α-CD** could also be kept constant for any desired duration of time. As a result, the ICD signal of **Mg-CP⊃α-CD** remains constant at a particular temperature, which is an inherent property of this host–guest complex. Therefore, by increasing or decreasing the temperature at regular intervals, a

definite amount of **α-CD** can be released or captured by the **Mg-CP** in a reversible and stepwise fashion (Fig. 6c). This property also allowed the release or capture of **α-CD** in different instalments by adjusting the temperature of the solution. This host–guest complex thus manifests itself to be programmable on an increasing or decreasing temperature scale. Any desired release or capture profile may be obtained by controlling the host–guest ratio at a particular temperature for any desired amount of time. The ICD signal was found to be constant at 20 °C, even after 1 h (Fig. 6d).

In order to study the feasibility of temperature-controlled release and capture of **α-CD** in the solid state, temperature-dependent changes in the ICD spectrum of **Mg-CP⊃α-CD** was investigated in a film prepared from an aqueous solution of **Mg-CP⊃α-CD** over a quartz substrate (Supplementary Fig. 9). The positive and negative ICD signals were found to be red shifted to 392 nm and 470 nm, respectively, due to the aggregation of **Mg-CP⊃α-CD** in the solid state. No detectable changes were observed in the ICD spectrum upon heating or cooling the sample, confirming that the rigid phase does not favour the dynamics of the host–guest self-assembly process and hence a fluid or soft phase is essential. For example, synthetic hydrogels efficiently mimic the salient features of extracellular matrices as well as help in the creation of three-dimensional cellular microenvironments[38,39]. These materials trap a large amount of water in the quasi-solid state[40] and behave as an ideal system to study

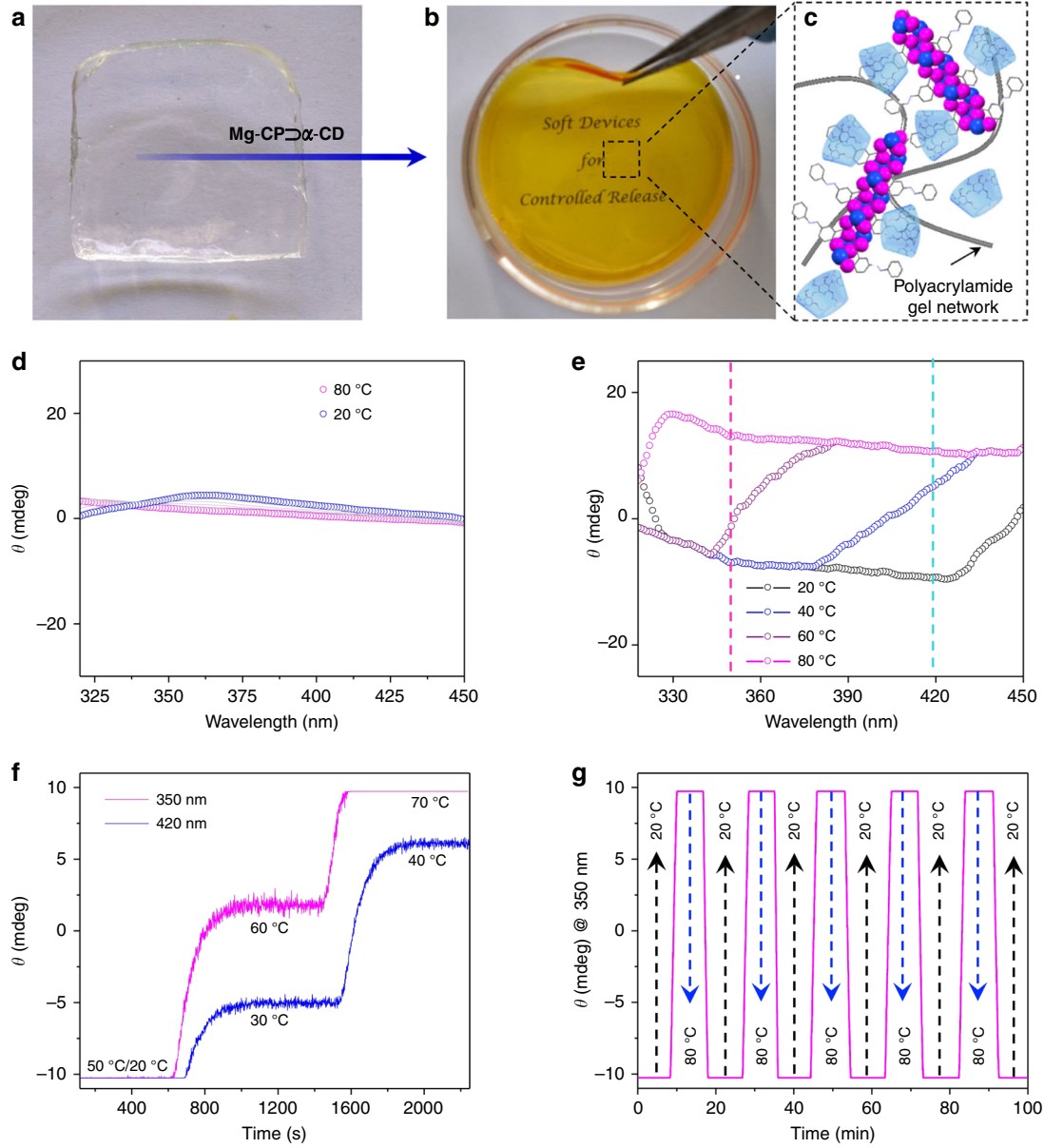

**Fig. 7** Stepwise and reversibly controlled release and capture of **α-CD** by **Mg-CP** in quasi-solid state. **a** Photograph of the polyacrylamide hydrogel. **b** Photograph of **Mg-CP⊃α-CD** incorporated into the polyacrylamide hydrogel. **c** A pictorial representation of **Mg-CP⊃α-CD** in the polyacrylamide hydrogel. **d** Temperature-dependent changes in the ICD spectra of the polyacrylamide hydrogel. **e** Temperature-dependent changes in the ICD spectra of the **Mg-CP⊃α-CD** entrapped polyacrylamide hydrogel. Changes in the ICD spectrum at 350 and 420 nm are represented by magenta and cyan lines, respectively. **f** Stepwise release profile of **α-CD** from **Mg-CP** monitored at 350 and 420 nm in the hydrogel at different temperatures. **g** Reversible binding (black arrow) and release (blue arrow) of **α-CD** from **Mg-CP** incorporated into the hydrogel

temperature-controlled host–guest interactions in a restricted environment. **Mg-CP⊃α-CD** was therefore incorporated into a polyacrylamide hydrogel (Fig. 7a–c). The yellow colour of the obtained soft material confirmed the incorporation of **Mg-CP⊃α-CD** within the gel matrix. The hybrid gel was found to be highly mouldable, flexible and transparent (Fig. 7b). No noticeable change in the CD spectrum of the pristine hydrogel was observed upon heating and cooling between 20 and 80 °C (Fig. 7d). Specifically, the CD intensity at 350 nm and 420 nm corresponding to **Mg-CP⊃α-CD** remained constant with respect to the changes in the temperature of the gel (Supplementary Fig. 10). The ICD spectrum of **Mg-CP⊃α-CD** incorporated inside the hydrogel matrix (Fig. 7e) recorded at different temperatures was different from that of the complex in solution (Fig. 5a). The

nature of the ICD spectrum is decided by a set of rules proposed by Harata and Kodaka initially for CD inclusion complexes and generalized later for chiral macrocycles[34]. These rules state that (i) any electronic transition that is polarized parallel or perpendicular to the axis of the macrocyclic host gives a positive and negative ICD signal, respectively; (ii) on movement of the chromophore from inside of the macrocycle cavity to outside, keeping the direction of the electronic transition moment unaltered, the sign of ICD is reversed; (iii) when a chromophore remains outside, just near the narrow rim of the CD, the intensity of the ICD signal is greater when compared to being positioned just outside of the wider rim; and (iv) the intensity of an ICD signal due to an electronic transition polarized perpendicular to the macrocycle axis is $-1/2$ of that when the transition is

polarized in the parallel direction and the sign of the ICD signal changes at 54.7°. In the solution state, according to rules (i) and (iv), **Mg-CP⊃α-CD** shows a positive ICD signal (due to π–π* transition) and a negative signal (due to n–π* transition). However, in the hydrogel matrix, the incorporated **Mg-CP⊃α-CD** shows a broad negative ICD signal, corresponding to both π–π* and n–π* transitions, in accordance with rules (i), (iii) and (iv). The changes may be attributed to the complex positioning of the **Mg-CP⊃α-CD** polymer strands and free **α-CD** close to one another in the gel, in a haphazard fashion, as opposed to the larger inter-strand distance in solution, where electronic interactions are minimal. In order to prove the stability of **Mg-CP⊃α-CD** inside the polymer hydrogel matrix, leaching of the complex into water from a xerogel strip at room temperature was monitored at 355 nm (Supplementary Fig. 11a). The ICD spectrum of the aqueous phase matched that of **Mg-CP⊃α-CD** complex in solution (Supplementary Fig. 11b).

On heating the gel sample, the ICD signal is completely reversed according to rule (ii). Temperature-dependent ICD spectrum of **Mg-CP⊃α-CD** trapped inside the hydrogel matrix was monitored at 350 and 420 nm (Fig. 7e and Supplementary Fig. 12), corresponding to the major ICP bands of **Mg-CP** in solution. Contrary to the solution-state ICD spectrum (Fig. 5b, inset), a clear hysteresis was observed between the heating and cooling cycles of **Mg-CP⊃α-CD** when incorporated into the polymer hydrogel. This observation indicates that, although **α-CD** molecules were released from the **Mg-CP** on heating, they were not readily available to recapture the free azobenzene sites on cooling. Temperature-dependent swelling and de-swelling of the polymer hydrogel promotes a lag phase between these binding and release processes, leading to the observed hysteresis, similar to the hysteresis loop in the gas adsorption profile of soft porous crystals[41]. Akin to the solution state, the ICD signal could be consistently modulated by maintaining the hydrogel at a particular temperature. A stepwise release profile of **α-CD** from the **Mg-CP** could also be demonstrated (Fig. 7f) as in solution. The temperature-dependent ICD spectrum of the polymer hydrogel, monitored at 350 and 420 nm (Supplementary Fig. 10), where no perceptible changes was observed, substantiates that the differences observed above originate solely from the host–guest interactions involving **α-CD** and **Mg-CP**. The system was also found to maintain its high reversibility for several heating and cooling cycles in the quasi-solid state (Fig. 7g).

## Discussion

Moving away from the conventional approach, in the present study, CDs have been used as guests for a CP **Mg-CP**. The water-soluble CP having CD-binding azobenzene-based side appendages possess a greater affinity and high selectivity towards various CDs as compared to its organic precursor. The **Mg-CP⊃α-CD** precisely perform quantitative capture–release process using temperature as stimulus. The chemistry discussed herein reaps the thermodynamic benefits of a system in chemical equilibrium and may have potential impact in designing systems for controlled delivery of stealth drug molecules, such as CDs.

## Methods

**Stepwise CD capture and release experiments**. Temperature-controlled stepwise capture and release of **α-CD** was performed in the time course measurement mode. Relative changes in ICD signal intensity was monitored at a particular wavelength (corresponding to π–π* transition) with respect to time. In all samples, 90 μL of **Mg-CP** (2 mg mL$^{-1}$) was added to a 3 mL solution of **α-CD** ($1 \times 10^{-3}$ M) in water. The temperature of the solution was manually changed using the thermostat between specific temperature ranges (e.g. from 80 to 70 °C → from 80 to 20 °C, Fig. 5c). Considering the maximum achieved ICD signal as 100%, the amount of **α-CD** captured was determined. The temperature of the thermostat was reduced manually between 90 and 20 °C, varying the rate from 5 to 15 °C min$^{-1}$, to obtain

Fig. 6a. The experiment was repeated at least three times to confirm our observation. When the solution was cooled and heated with a particular rate, in repeated cycles, Fig. 6b was obtained. When the temperature of the solvent was kept constant for a particular time interval (5 min) during the cooling or heating process, the ICD signal versus time showed a plateau-like behaviour (Fig. 6c). For the experiments in the hydrogel state, a hydrogel/**Mg-CP/α-CD** composite was prepared inside a quartz quvette by adding 750 μL of an aqueous solution of **Mg-CP** (0.26 mg mL$^{-1}$) to 750 μL of an aqueous solution of **α-CD** (4.33 mg mL$^{-1}$). To the resulting solution, 1.5 mL of 40% stock solution of acrylamide/bisacrylamide was added followed by 300 μL of 10% ammonium persulphate solution. The resulting mixture was vigorously whisked to prevent the formation of air bubbles. Subsequently, 50 μL of tetramethylethylenediamine solution was added to control the rate of the polymerization process. A total of 3 mL of the above solution was left undisturbed for ~1 h in a quartz cuvette to obtain a transparent pale yellow hydrogel. The circular dichroism spectral intensities at required constant temperatures were measured and plotted against time for monitoring the CD capture–release processes.

**Data availability**. All relevant data are available in the article as well as in the supplementary information files and from the authors on reasonable request.

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

## Acknowledgements

A.A. is grateful to Department of Science and Technology (DST), Government of India for a J. C. Bose Fellowship (SB/S2/JCB-11/2014). R.D.M. and G.D. are thankful to Council of Scientific and Industrial Research (CSIR), Government of India for research fellowships. We thank Mr. Kiran Mohan, Dr. J. D. Sudha and Dr. E. Bhoje Gowd of CSIR-NIIST for TEM, rheology and WAXS measurements, respectively.

## Author contributions

R.D.M. undertook the synthesis of molecules and performed studies. G.D. helped in ICD measurements and hydrogel preparation. A.A. and R.D.M. designed the experiments. All the authors analysed the data, discussed the results, wrote and commented on the manuscript. A.A. was responsible for the overall project concept, direction and coordination.

## Additional information

**Competing interests:** The authors declare no competing interests.

