## [Peer Review File · Nature Communications]

Reviewers' comments:

Reviewer #1 (Remarks to the Author):

The paper by Ajayaghosh and co-workers describes the stepwise release and capture of cyclodextrin (CD) guests from a supramolecular gel using temperature as stimulus. The system is composed of a coordination polymer that, undergoing lateral complexation with CD, operates as a fully reversible material. Usually, temperature is not a suitable stimulus for triggering artificial host-guest interaction as it typically leads to continuum releases. In this work, however the authors describe a unique prototype manifesting a temperature-controlled natural host-guest interaction. Specifically, this coordination polymer, when incorporated in a hydrogel matrix, lead to a microenvironment that facilitates the stepwise release of α -CD responding to a temperature variation. The paper is very nice and well written. The scientific achievements are very clear and strongly supported by a coherent experimental approach. I thus strongly support its publication. Only a couple of suggestions:

- 1) The authors should provide a figure (cartoon-like) that describes the concepts that they want to put forward in general terms. This will certainly help any readers to better understand the content and increasing its impact in the wide scientific community.
- 2) The authors should also cite the following the papers describing thermo-responsive gels: Adv. Mater. 2002, 14, 1120 and Adv. Mater. 2013, 15, 2462.

Reviewer #2 (Remarks to the Author):

Ajayaghosh and co-workers report temperature control of host guest interactions in order to afford the stepwise release and capture of cyclodextrins with a coordination polymer. They present detailed studies ranging from the coordination polymer preparation, gelation of the Mg-CP polymer, and loading as well as release of the cyclodextrins using CD analysis. While the study described here is systematic, the concepts claimed were not adequately supported. The use of temperature to control host-guest complexation does not constitute "precise" release or capture, rather it results in a continuous shift in equilibrium, which has been shown in many previous reports of CD bound to a suitable guest pendent from polymer chains or on a surface. It recommended that a complete re-framing of the manuscript is required before it can be re-evaluated in a more targeted journal.

Several comments that the authors should take into account when re-framing their manuscript:

I am not fully convinced by the "precise release" concept the authors suggest. Host-guest complexation is strongly dependant on temperature. It is not surprising that an increase in temperature results in an increase the binding dynamics, leading to changes in the CD signals. Moreover, cooling of the system back to its original temperature will favour complex formation. What the authors observe is rather the use of temperature to manipulate the relative on-off rates rather than precisely controlling release.

Has any biophysical characterization been performed on the Mg-CP@CD complexes? The authors claim in the introduction that "CDs have also been recognized as potent drug molecules for treating fatal neurodegenerative disorders like Niemann-Pick type C and as anti-obesity dietary fibers for flushing out fat from high cholesterol diets". In order to substantiate such claims, a simple in vitro experiment should be included to confirm that the temperature-induced "release" of CD can really function in such a manner.

As the authors include the use of an azobene derivative, why is photo-irradiation not utilized to trigger complexation/decomplexation rather than or in addition to temperature.

How is the stability of the Mg-CP in the polyacrylamide hydrogel? Is it still stable as a polymer, or it dissociates into small oligomers?

It is not clear how the critical gelation concentration was determined in these studies (mentioned on page 5 lines 82-83). A quantitative measurement is required.

On page 8 lines 109-110, it is mentioned that the ICD signal is found to be 2x higher for the Mg-CP complexed with 3.5 eq of alpha-cyclodextrin compared to AzSA complexed with 4.5 eq alpha-cyclodextrin. It would be interesting to see how the Mg-CP and AzSA bind with cyclodextrins of different size (e.g. alpha vs beta).

Reviewer #3 (Remarks to the Author):

This manuscript by Ajayagosh and co-workers reports a coordination polymer (Mg-CP) with pendant azobenzene units that is capable of reversibly binding and releasing alpha-cyclodextrin moieties in aqueous solution driven by host-guest interactions between the azo groups and the CD units. Interestingly, the coordination polymer is able to capture and release alpha-CD in a stepwise and controlled fashion by simply controlling the ratio of CP and CD units and the temperature. The CP was ultimately incorporated into a hydrogel matrix and similarly to the solution behaviour, a stepwise release of alpha-CD units in response to temperature could be demonstrated.

Novelty, significance and relevance of the work: As the authors mention in the introduction, CDs are typical host molecules for a variety of hydrophobic molecules in aqueous media. In this regard, the present work is a rare example where CD acts as a guest rather than a host, justifying the elegant and novel approach of the authors. This aspect will definitely influence thinking in the field and encourage other scientists to use CDs as guest units. Additionally, due to the inherent properties and key implications of CDs in multiple research fields such as biomedicine and polymeric materials, designing efficient systems to capture and release them in a controlled fashion is a very important issue that the authors have successfully tackled in the current work. Thus, this piece of work will be of high interest to many other scientists in the field of supramolecular chemistry, aqueous self-assembly and host-guest systems. The manuscript has been clearly written. Finally, the technical quality of the manuscript is very high. All experiments have been performed with care and the conclusions are supported by the experimental data.

Based on the above, I recommend this paper for publication in Nature Communications after the authors have addressed the following points:

- 1) The host-guest pair CD-azobenzene has been widely investigated in the literature. This interaction is often disturbed upon photoisomerization of the azobenzene unit in the presence of UV light. For instance, numerous examples of hydrogels based on host-guest interactions CD-azobenzene are known to disassemble/reassemble upon UV/visible light irradiation. In this regard, the authors have used exactly this pair of host/guest systems but have not investigated their light-responsive behavior. I anticipate that the future readers of this work would ask the same question: are the present systems light-responsive? How do the systems behave in terms of binding/release of CD in the presence of light? In other words, it needs to be addressed why the authors have chosen azobenzene for their molecular design and not stilbene or other hydrophobic units as binders for alpha-CD.
- 2) On page 7 the authors mention that the ligand AzSA does not show any ordering in WAXS experiments. I suggest the authors to explain this more in detail and assign the peaks of AzSA in Figure 2 to a pattern.
- 3) The authors also mention the "stepwise" capture and release of CD by the CP. After reading the manuscript carefully for a couple of times, I personally could not follow completely how the authors perform these experiments, see for instance figure 4c or figures 5a,c. I recommend the

authors to describe their procedure in the main text more in detail, i.e. how exactly time, temperature and ratio/concentration of host and guest are changed. What is the concentration and ratio of host and guest? How do the curves shown in figures 4c, 5a and 5b arise? Please also incorporate this information to the caption of figures 4-5.

Point-by-Point Answers to the Reviewer's Comments

Reviewer # 1

The paper by Ajayaghosh and co-workers describes the stepwise release and capture of cyclodextrin (CD) guests from a supramolecular gel using temperature as stimulus. The system is composed of a coordination polymer that, undergoing lateral complexation with CD, operates as a fully reversible material. Usually, temperature is not a suitable stimulus for triggering artificial host-guest interaction as it typically leads to continuum releases. In this work, however the authors describe a unique prototype manifesting a temperature-controlled natural host-guest interaction. Specifically, this coordination polymer, when incorporated in a hydrogel matrix, lead to a microenvironment that facilitates the stepwise release of α -CD responding to a temperature variation. The paper is very nice and well written. The scientific achievements are very clear and strongly supported by a coherent experimental approach. I thus strongly support its publication. Only a couple of suggestions:

(i) Comment: The authors should provide a figure (cartoon-like) that describes the concepts that they want to put forward in general terms. This will certainly help any readers to better understand the content and increasing its impact in the wide scientific community.

We thank the reviewer for the valuable suggestion. We have added a cartoon to introduce the concept to the readers in the revised manuscript (Figure 1 in the revised manuscript, Page 2).

(ii) Comment: The authors should also cite the following the papers describing thermo-responsive gels: Adv. Mater. 2002, 14, 1120 and Adv. Mater. 2013, 15, 2462

The suggested references have been cited (References 16 and 17 in the revised manuscript).

Reviewer # 2

Ajayaghosh and co-workers report temperature control of host guest interactions in order to afford the stepwise release and capture of cyclodextrins with a coordination polymer. They present detailed studies ranging from the coordination polymer preparation, gelation of the Mg-CP polymer, and loading as well as release of the cyclodextrins using CD analysis. While the study described here is systematic, the concepts claimed were not adequately supported. The use of temperature to control host-guest complexation does not constitute "precise" release or capture, rather it results in a continuous shift in equilibrium, which has been shown in many previous reports of CD bound to a suitable guest pendent from polymer chains or on a surface. It recommended that a complete re-framing of the manuscript is required before it can be re-evaluated in a more targeted journal.

(i) Comment: I am not fully convinced by the "precise release" concept the authors suggest. Host-guest complexation is strongly dependant on temperature. It is not surprising that an increase in temperature results in an increase the binding dynamics, leading to changes in the CD signals.

Moreover, cooling of the system back to its original temperature will favour complex formation. What the authors observe is rather the use of temperature to manipulate the relative on-off rates rather than precisely controlling release.

We partially agree with the reviewer that the temperature dependent host-guest complexation results in a continuous shift of equilibrium. However, herein, we have reported a rather step-wise change in the host-guest complexation ratio by providing a certain amount of heat energy to the complex in solution (rather than on-off), which according to our knowledge, has not been demonstrated so far. However, in order to avoid any confusion in this regard we have modified the title to ‘Stepwise control of host-guest interaction using a coordination polymer gel’.

(ii) Comment: *Has any biophysical characterization been performed on the Mg-CP@CD complexes? The authors claim in the introduction that “CDs have also been recognized as potent drug molecules for treating fatal neurodegenerative disorders like Niemann-Pick type C and as anti-obesity dietary fibers for flushing out fat from high cholesterol diets”. In order to substantiate such claims, a simple in vitro experiment should be included to confirm that the temperature-induced “release” of CD can really function in such a manner.*

We thank the reviewer for the valuable suggestion. As we have mentioned in this manuscript that cyclodextrins (CDs) have been mostly utilized in the scientific literature, as a supramolecular host. Herein, we report a rare example of CD being used as a guest. In the mentioned lines, we wanted to highlight the importance of CD as a guest (as a drug or as a supramolecular synthon). However, demonstration of such a drug delivery system is beyond the scope of the present manuscript. It may be possible to develop a drug delivery system utilizing the same principle but with a different substrate design.

(iii) Comment: *As the authors include the use of an azobenzene derivative, why is photo-irradiation not utilized to trigger complexation/decomplexation rather than or in addition to temperature.*

We would like to inform the reviewers that we have initially carried out photoirradiation experiments, but we did not observe any change in the UV-Vis absorption spectrum of Mg-CP even on prolonged UV irradiation. It is well known in the scientific literature that coordination of an azobenzene ligand, in a coordination polymer or MOF backbone may lead to the loss of its photoisomerisation properties (Ref 30-32 in the revised manuscript). Additionally, we assume that due to the presence of an electron donating hydroxy group at the *para* position to the azobenzene moiety, the system may undergo very fast *cis-trans* reverse isomerization (Ref 33 in the revised manuscript). As a result, spectroscopic changes were not observed in the experimental time scale. The details of the photoisomerisation experiment are now added in the revised manuscript (Page 6) and Supplementary Information (Page 4 and Supplementary Figure 2).

(iv) Comment: How is the stability of the Mg-CP in the polyacrylamide hydrogel? Is it still stable as a polymer, or it dissociates into small oligomers?

We have observed that **Mg-CP** can still bind to α -CD molecules once released outside the gel (Supplementary Figure 11), confirming the molecular stability of **Mg-CP**.

(v) Comment: It is not clear how the critical gelation concentration was determined in these studies (mentioned on page 5 lines 82-83). A quantitative measurement is required.

The critical gelation concentration in case of a coordination polymer based gelator is usually reported based on the minimum concentration of any one of the components required to synthesise a nonflowing gel in a particular volume of the solvent. In this case, we have varied the concentration of the organic linker (**AzSA**) in 1 mL of the solvent (DEF). The details of the experiment are provided in the Supplementary Information (Page 2) of the revised manuscript.

(vi) Comment: On page 8 lines 109-110, it is mentioned that the ICD signal is found to be 2x higher for the Mg-CP complexed with 3.5 eq of alpha-cyclodextrin compared to AzSA complexed with 4.5 eq alpha-cyclodextrin. It would be interesting to see how the Mg-CP and AzSA bind with cyclodextrins of different size (e.g. alpha vs beta).

We thank the reviewer for the valuable suggestion. We have observed that **Mg-CP** shows different behaviour with cyclodextrins with different size like α -CD and β -CD. The chemical stimuli-responsive behaviour of this system will be communicated in the near future. Research in this direction is currently under progress.

Reviewer # 3

This manuscript by Ajayagosh and co-workers reports a coordination polymer (Mg-CP) with pendant azobenzene units that is capable of reversibly binding and releasing alpha-cyclodextrin moieties in aqueous solution driven by host-guest interactions between the azo groups and the CD units. Interestingly, the coordination polymer is able to capture and release alpha-CD in a stepwise and controlled fashion by simply controlling the ratio of CP and CD units and the temperature. The CP was ultimately incorporated into a hydrogel matrix and similarly to the solution behaviour, a stepwise release of alpha-CD units in response to temperature could be demonstrated.

Novelty, significance and relevance of the work: As the authors mention in the introduction, CDs are typical host molecules for a variety of hydrophobic molecules in aqueous media. In this regard, the present work is a rare example where CD acts as a guest rather than a host, justifying the elegant and novel approach of the authors. This aspect will definitely influence thinking in the field and encourage other scientists to use CDs as guest units. Additionally, due to the inherent properties

and key implications of CDs in multiple research fields such as biomedicine and polymeric materials, designing efficient systems to capture and release them in a controlled fashion is a very important issue that the authors have successfully tackled in the current work. Thus, this piece of work will be of high interest to many other scientists in the field of supramolecular chemistry, aqueous self-assembly and host-guest systems. The manuscript has been clearly written. Finally, the technical quality of the manuscript is very high. All experiments have been performed with care and the conclusions are supported by the experimental data.

(i) Comment: The host-guest pair CD-azobenzene has been widely investigated in the literature. This interaction is often disturbed upon photoisomerization of the azobenzene unit in the presence of UV light. For instance, numerous examples of hydrogels based on host-guest interactions CD-azobenzene are known to disassemble/reassemble upon UV/visible light irradiation. In this regard, the authors have used exactly this pair of host/guest systems but have not investigated their light-responsive behavior. I anticipate that the future readers of this work would ask the same question: are the present systems light-responsive? How do the systems behave in terms of binding/release of CD in the presence of light? In other words, it needs to be addressed why the authors have chosen azobenzene for their molecular design and not stilbene or other hydrophobic units as binders for alpha-CD.

We thank the reviewer for the encouraging words. We would like to mention that although we have exploited the azobenzene chromophore in this work, as per the reviewer's suggestions other chromophores like functionalized stilbenes and even small molecules can also be utilized to develop thermal stimuli-responsive coordination polymer backbone. For details related to photoisomerisation, kindly see answer to Reviewer # 2, Comment (iii).

(ii) Comment: On page 7 the authors mention that the ligand AzSA does not show any ordering in WAXS experiments. I suggest the authors to explain this more in detail and assign the peaks of AzSA in Figure 2 to a pattern.

As per the reviewer's suggestion, the WAXS pattern of **AzSA** has been assigned to a particular structure (Page 7 and Supplementary Figure 3).

(iii) Comment: The authors also mention the "stepwise" capture and release of CD by the CP. After reading the manuscript carefully for a couple of times, I personally could not follow completely how the authors perform these experiments, see for instance figure 4c or figures 5a,c. I recommend the authors to describe their procedure in the main text more in detail, i.e. how exactly time, temperature and ratio/concentration of host and guest are changed. What is the concentration and ratio of host and guest? How do the curves shown in figures 4c, 5a and 5b arise? Please also incorporate this information to the caption of figures 4-5.

We thank the reviewer for the valuable suggestion. We have added appropriate experimental details in the revised manuscript.

Reviewers' comments:

Reviewer #2 (Remarks to the Author):

Although the system described in this work is well characterized, the property of stepwise control emphasized here is trivial and even misleading.

1. The stepwise profiles shown in this manuscript arise solely from the control of temperature in a stepwise manner. One can readily produce any of the stepwise profiles seen in Fig 5 or 6 by monitoring a temperature-dependent physical property of a thermodynamic equilibrated system, e.g. the viscosity of a liquid.

2. The signal change from the induced circular dichroism (ICD) is offered as the only indicator of what the authors term as "capture and release" of cyclodextrins, however, it is not convincing evidence. The change in ICD observed can be alternatively explained by a conformational change on account of temperature variance. The authors should design and present a dialysis experiment in order to demonstrate a true 'capture' and 'release' event.

Therefore, although this manuscript is well written, the novelty and scientific impact is not adequate for publication in Nature Communications.

Reviewer #3 (Remarks to the Author):

The authors have properly addressed the comments of all Referees and the manuscript is now recommended for final acceptance in its current form.

List of Additional Figures

Not added

Summary of Revisions

No new revisions included

Point-by-Point Answers to the Reviewer's Comments

Reviewer 2

(i) Comment: The stepwise profiles shown in this manuscript arise solely from the control of temperature in a stepwise manner. One can readily produce any of the stepwise profiles seen in Fig 5 or 6 by monitoring a temperature-dependent physical property of a thermodynamic equilibrated system, e.g. the viscosity of a liquid.

Yes, it is true that the stepwise control is achieved by control of temperature, which is the stimulus for the release-capture processes. In our case, we could achieve the stepwise control of the capture-release process not only by controlling the temperature but also by choosing the coordination polymer as the host and CD as the guest (usually CD act as the host), which is the novelty of the present work. It is not appropriate to compare the changes of a host-guest complex with a property like viscosity.

(ii) Comment: The signal change from the induced circular dichroism (ICD) is offered as the only indicator of what the authors term as "capture and release" of cyclodextrins, however, it is not convincing evidence. The change in ICD observed can be alternatively explained by a conformational change on account of temperature variance. The authors should design and present a dialysis experiment in order to demonstrate a true 'capture' and 'release' event.

We politely disagree with the reviewer on this comment. Circular dichroism or induced circular dichroism are powerful tools to study host-guest complexation, particularly with cyclodextrin. From Supplementary Fig. 6a, it is quite clear that both **AzSA** and **Mg-CP** do not exhibit any ICD signal in the absence of cyclodextrin. With the gradual addition of cyclodextrin, the ICD signal is slowly increased (Supplementary Fig. 6b). The ICD signal therefore solely comes from the interaction of cyclodextrins with **Mg-CP** or **AzSa**. The signal decreases with a decrease in such interaction (on increasing temperature). Changing the temperature of a solution of **AzSA** or **Mg-CP** alone therefore does not give any change in the ICD signal (with respect to conformational changes, if occurred any). Conformational changes to the coordination polymer does not bring any changes in the ICD signal.

The dialysis experiment as suggested by the reviewer, to completely separate **Mg-CP** from cyclodextrin, is difficult and possibly improbable with our system. Also, cyclodextrin cannot be detected alone by ICD. A temperature dependent dialysis experiment therefore looks clearly outside the scope of the present work. However, we will keep in mind the suggestion of the reviewer for our studies with other systems.

Reviewer 3

The authors have properly addressed the comments of all Referees and the manuscript is now recommended for final acceptance in its current form.

We thank the reviewer for carefully going through our revised manuscript and accepting it in its present form.

REVIEWERS' COMMENTS:

Reviewer #3 only made comments to the editor.

Reviewer #3 looked over the authors' response to reviewer #2's comments in the last round of review and was satisfied with the response.